# The Net Spatio-Temporal Impact of the International Tourism Is-Land Strategy on the Ecosystem Service Value of Hainan Island: A Counterfactual Analysis

**Miao Guan and Changsheng Xiong ***

Department of Land Resources Management, School of Public Administration, Hainan University, Haikou 570228, China
* Correspondence: xiongcs@hainanu.edu.cn

**Abstract:** The strategy of building an international tourism island in Hainan is an important national strategic deployment, with tourism as the core, integrating a series of issues such as industry, tropical agriculture, and urban-rural relations. The implementation of this strategy profoundly affects the evolution of local land use patterns and ecosystems on Hainan Island. This paper utilizes a counterfactual analysis framework and Cellular Automata (CA)-Markov model based on the current land use data of Hainan Island for the three periods of 1999, 2008 and 2017. Accordingly, the spatial and temporal conditions of ecosystem service values (ESV) in 2017, under the assumed scenario of unimplemented international tourism island strategy, were simulated. The net spatial and temporal impacts of the international tourism island construction strategy on the value of ecosystem services on Hainan Island were finally assessed. The results are as follows. First, the total value of ESV in Hainan Island in 1999, 2008 and 2017 were 33.88 billion yuan, 56.045 billion yuan and 50.417 billion yuan respectively showing a trend of first increasing and then decreasing; spatially, the ESV were high in the central region and low in the surrounding areas. Second, in the simulated scenario without the implementation of the international tourism island construction strategy in 2017, the total ESV of Hainan Island was 54.19 billion yuan. Third, the implementation of the international tourism island policy reduced the ESV by 3773 million yuan, and the impact of this policy was high in coastal areas and low inland. There was an obvious divergence between the positive and negative effects.

**Keywords:** international tourism island construction strategy; value of ecosystem services; counterfactual analysis framework; spatial and temporal net impacts; Hainan

## 1. Introduction

From coastal special economic zones to western development to the construction of free trade zones (ports), China has introduced a series of regional development strategies [1] that have become a major feature of the urbanization and industrialization process [2]. Regional development strategies are plans for overall development that encompass regional economic, social, political, cultural, and ecological aspects [3], and their implementation indicates the direction of and goals for regional development and provides necessary policy support, which is an important driving force for regional economic and social development.

However, the implementation of regional development strategies has also had negative impacts on ecosystems, such as forest encroachment, land degradation, water loss, and biodiversity reduction, which cannot be ignored [4], leading to the weakening of ecosystem service functions and the reduction of ecological values. In the context of the emergence of regional development strategies and the continuous promotion of ecological civilization in China, the coordination of the relationship between strategy implementation and ecosystem service functions has become an important issue of concern. An accurate understanding of the performance of regional development strategies on ecosystem service functions is an important foundation for current research and the main research of this paper.

## 2. Review of the Literature

To capture the impact of regional development strategies on ecosystem service functions, the first step is to quantify the ecosystem service function. Currently, quantitative assessment of ecosystem service functions is mainly achieved by accounting for ecosystem service values (ESVs) [5]. Ecosystem service functions are the direct or indirect benefits that humans currently derive from ecosystems to meet their needs, including provisioning, regulating, supporting, and recreational services [6]. ESVs, on the other hand, are a market-based or near-market-based valuation of the different service functions provided by ecosystems, as a reflection of the level of ecosystem service functions [7].

There are two main methods to account for the value of ecosystem services. One is the functional value approach, based on the price per unit of service function [8]. This method is based on economics; it establishes the production equation between a certain service function and local ecological variables, and simulates the calculation to obtain the value of ecosystem services in the region [9,10]. However, the process of executing this method is tedious, and it is difficult to make a breakthrough in specific theories and methods at this stage. The second approach is the equivalent factor method based on the value of unit area [11]. This method uses quantifiable criteria to construct the value equivalents of different ecosystem types and different service functions. It combines the area size for ESV assessment, which is more intuitive and easier to use [12–19]. There are also studies that have tried to revise the approach to accounting for ESVs. For example, Xie et al. [20] introduced net primary productivity (NPP) spatial and temporal adjustment factors, precipitation spatial and temporal adjustment factors, and soil conservation spatial and temporal adjustment factors as a way to distinguish the variability of ESVs in different provinces in different months.

On the other hand, it is also necessary to capture and assess the ecological effects of regional development strategies. From the results of the effects, there are three main types of impacts of regional development strategies on ecosystems. First, policy implementation affects regional land use/cover and thus changes with the ecological functions of land-use types [21–25]. For example, van Meijl et al. [26] analyzed the effects of different policy environments on agricultural land in Europe, and Wu [27] and others elaborated the effects of urbanization policies on land-use changes.

Second, policy implementation affects the ecological security pattern of regional landscapes [28,29]. For example, Ding et al. [30] and others took the south-central region as an example to construct the ecological security pattern of the rapid urbanization strategy area and proposed an optimization strategy to obtain a more reasonable ecological corridor direction for the construction of the ecological security pattern. Tyler et al. [31] analyzed the impact of forest policy on the late forest landscape pattern of the Olympic Peninsula in western Washington State.

Third, policy implementation at the biological level directly interferes with ecological cycles within ecosystems and has an impact on them [32,33]. For example, rice—one of the most important agricultural products—is grown to ensure food security, and policies are enacted in almost all countries in the world to support this goal. Miranda et al. [34] used this concept to describe the environmental elements, greenhouse gas production mechanisms, and ecological restoration associated with rice cultivation in an attempt to mitigate global warming.

In terms of assessing the performance of regional development strategies on ecological impacts, there are three main approaches: before–after, control intervention, and the before–after and control intervention (BACI) assessment framework, which combines the former two methods [35–40]. The before–after assessment framework refers to the comparative analysis of scenarios before and after the implementation of the strategy to derive the spatial and temporal impact of the strategy in the study area [41]. The control intervention assessment framework describes the comparison of the implementation area of the strategy with the non-implementation area to derive the impact of the strategy from the salient characteristics of the intervention group [42]. The BACI assessment framework is based on

the comparison of different implementation scenarios of the same area [40]. For example, Wauchope et al. [43] compared and synthesized these two analytical frameworks and eventually explored the impact of 1506 protected areas on the global population trajectory of 27,055 waterbirds using a robust before-after control intervention model, providing recommendations for international ecological policy and management.

The above studies are important references for assessing the impact of regional development strategies on the value of ecosystem services, but they are based on the before–after assessment framework, which ignores the superimposed effects of other factors (such as cities' own development) during the same period. Moreover, the control intervention framework faces difficulties in selecting control groups. The BACI framework is based on the control intervention approach, which exhibits challenges in selecting the control group and cannot effectively exclude the special attributes of the intervention and control groups in terms of geographic location, development environment, and economic base. Furthermore, the BACI framework only uses the before–after and control intervention frameworks to conduct mutual robustness tests, which does not fundamentally address the limitations faced by each of the two assessment frameworks.

To address the above issues, this research takes the 2008 Hainan International Tourism Island construction strategy (hereafter referred to as "ITIS strategy") as an example and introduces a counterfactual analysis framework to assess its impact on the ESV of Hainan Island. The net spatio-temporal impact of the ITIS is assessed. There are several main reasons for the selection of the ITIS strategy for this study. First, the implementation of the ITIS provides better research material for this paper. After nearly a decade of development since Hainan Province proposed the "Action Plan for the Construction of Hainan International Tourism Island" in 2008 and the Chinese State Council promulgated the "Opinions on Promoting the Construction and Development of Hainan International Tourism Island" in 2010, it is worth paying attention to how the implementation of these policies has affected the ESV of Hainan Island. Second, Hainan Island's overall socio-economic development level is low and is affected by the implementation of the ITIS. Third, as an independent geographical unit, Hainan Island's self-regulation capacity and resistance to external disturbances is extremely fragile; therefore, blind development may cause serious damage to the ecological environment of the island. The focus on the impact of the strategy on the ecosystem has important implications for the current construction of Hainan's free trade port and ecological protection.

Specifically, the spatial and temporal evolution of ESV on Hainan Island is obtained based on the historical land-use cover data of Hainan Island. The net spatial and temporal impact of strategy implementation on ESV is assessed by comparing the spatial and temporal distribution with the spatial and temporal distribution of ESV after the implementation of the ITIS strategy. Next, the counterfactual analysis framework and Cellular Automata (CA)-Markov model are combined to dynamically simulate the spatio-temporal distribution of ESV in Hainan Island under the hypothetical scenario that the ITIS strategy is not implemented, and to compare the analysis with the spatio-temporal distribution of ESV after the implementation of the ITIS strategy to assess the net spatio-temporal impact of strategy implementation on ESV.

The rest of the paper is organized as follows: Section 2 introduces the assessment ideas based on the counterfactual analysis framework; Section 3 presents the data and methods; Section 4 empirically assesses the results of the spatial and temporal impacts of the ITIS on the value of ecosystem services on Hainan Island; Section 5 presents a discussion of the assessment results; and the last section concludes.

## 3. Assessment Ideas Based on a Counterfactual Analysis Framework

To assess the spatio-temporal impact of the ITIS on the ESV of Hainan Island, the traditional method of assessing the change differential is based on a before–after analysis framework by comparing the differences in the spatio-temporal distribution states of ESV before and after the implementation of the strategy (Figure 1a). However, this assessment

idea ignores the role of confounders in the policy, making the obtained conclusions inaccurate. In other words, it ignores the fact that land use and the ecosystem of Hainan Island will not stop evolving even if the strategy is not implemented. Therefore, the assessment results are likely to be overestimated (i.e., they do not exclude the changes in ESV brought about by Hainan's own socio-economic development).

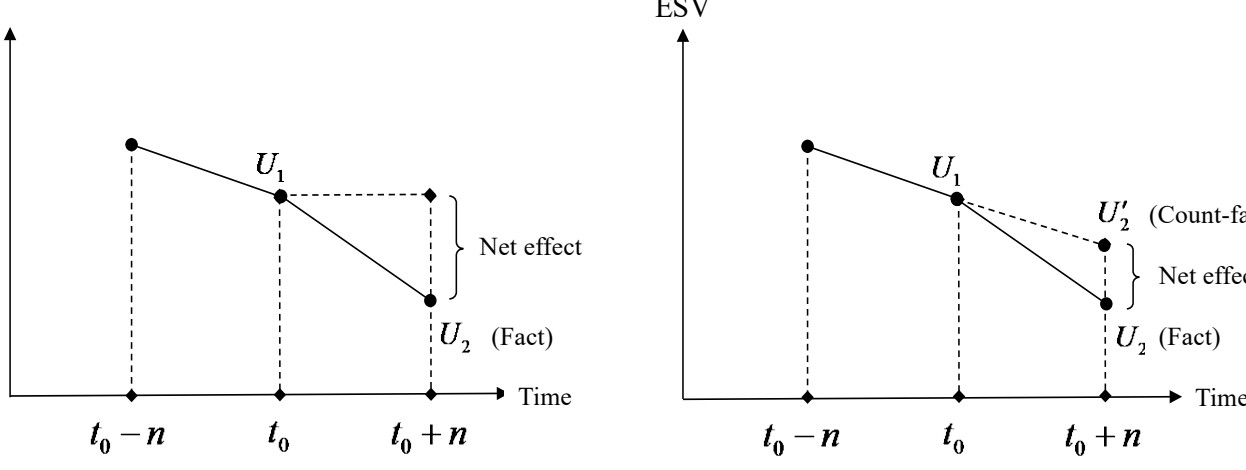

**Figure 1.** Schematic assessment of the impact of the implementation of the international tourism island construction strategy on the ESV of Hainan Island.

In this paper, drawing on the counterfactual analysis framework proposed by Rubin in 1974 [44], we consider the state of the ESV distribution at the moment $t_0 + n$ after the ITIS has been implemented as a fact (denoted as $U_2$); the state of the ESV distribution at the moment $t_0 + n$ after the ITIS is assumed to be unimplemented as a counterfactual (denoted as $U_2'$). The difference between the fact and the counterfactual $(U_2' - U_2)$ is the treatment effect of the ITIS implementation on ESV (Figure 1b). This assessment idea regards the difference between the predicted outcome under the counterfactual and the actual outcome under the fact as the treatment effect of policy implementation, which fully accounts for the changes in ESV brought about by regional socio-economic development under the scenario of the unimplemented ITIS, and can reveal the net impact of ITIS on the ESV of Hainan Island more objectively and effectively.

In the assessment process based on the counterfactual analysis framework, the key is to obtain the ESV distribution status of Hainan Island under the counterfactual scenario. Given that land-use cover change is an evolutionary process with typical Markovian properties [45–48], the land-use cover state at the current moment is only related to the land-use cover state at the previous moment, and the land-use cover change at the next moment is mainly influenced by the land-use cover state at the previous moment [49]. Therefore, it can be argued that under the counterfactual scenario where the ITIS is assumed not to be implemented, the ESV changes caused by land-use cover will largely follow the land-use cover evolution trend before the implementation of the ITIS. That is, the ESV distribution state $t_0 - n$ under the counterfactual scenario ($U_0$) can be simulated by the evolutionary trend of the ESV distribution state $t_0$ at the moment of ($U_1$) before the implementation of the ITIS to the ESV distribution state ($U_2'$) at the moment of the initial implementation of the strategy.

## 4. Data and Methods

### 4.1. Overview of the Study Area

Hainan Island, as the second largest island in China with 18 cities and counties under its jurisdiction, including Haikou and Sanya, is the main part of the land area of Hainan Province and the area covered by the implementation of the ITIS. Since the official implementation of the ITIS in 2008, Hainan Province has achieved significant social and economic development. By the end of 2017, the total residential population of the province was 9,257,600, the total GDP reached 446,254 billion yuan, and the structure of the three industries (agriculture, industry, and services) also changed from 37.44:20.15:42.41 to 21.6:22.3:56.1 with the tertiary industry, with tourism and real estate industry as the pillars, gaining rapid development. At the same time, driven by favorable policies and socio-economic development, the province's land-use structure is undergoing dramatic changes and driving the spatial and temporal evolution of ESVs.

This paper selects 1999, 2008, and 2017 as the research time points, with 1999–2008 reflecting the situation before the implementation of the ITIS and 2008–2017 characterizing the situation during and after the implementation of the ITIS. The year 2017 is taken as the study termination point to exclude the possible impact of the implementation of the strategy for the construction of China's (Hainan's) Pilot Free Trade Zone (Port) in 2018.

### 4.2. Data Sources

The data used in this paper involve both land-use and socio-economic data. The land-use data were obtained from the national historical CLCD dataset ("30 m Annual Land Cover and Its Dynamics in China from 1990 to 2019") by the research team of Xin Huang and Jiayi Li at Wuhan University based on the historical series of Landsat images and using the random forest classifier. The above data were pre-processed with ENVI software for radiometric calibration, atmospheric correction, and geometric correction. They were compared with high-definition remote sensing images and corrected with the actual situation of the study area. The socio-economic data mainly include the output value of food crops and Engel's coefficient, among other measures, which were obtained from the statistical yearbook or bulletin of Hainan Province.

### 4.3. CA-Markov Model

The Markov component of the CA-Markov model focuses on the prediction of the quantity of land-use change, but it cannot be spatially expressed, nor can it show the spatial distribution of each type of land change [50]. The meta-automata model, on the other hand, is able to express the spatio-temporal dynamic evolution process of complex spatial systems to compensate for the deficiency of the Markov model [51]. Simulating the evolution of land use using the CA-Markov model has good permeability and applicability [50]. The main working principle is based on the transformation area matrix, the transition probability matrix, and the set of suitability maps from the initial period (t − 1) to the base period (t), and the land use. The land use is reallocated to predict future land use.

In this paper, firstly, we input the land use data in 1990 and 1999 to generate the land use type transfer matrix and the probability phase diagram of land use type distribution in 1990–1999, so as to simulate the data set of land use distribution in 2008. Secondly, the simulated land use data in 2008 and the actual land use data in 2008 were compared and checked for accuracy. If the kappa coefficient reaches the accuracy standard, the next step of prediction can be carried out; if the kappa coefficient is lower than the accuracy standard, the relevant parameters are adjusted and the above steps are repeated until the accuracy reaches the standard. Finally, the land use data of 1999 and 2008 were input and the same steps as above were used to simulate the land use of Hainan Island in 2017 under the scenario of not implementing the international tourism island strategy (Figure 2).

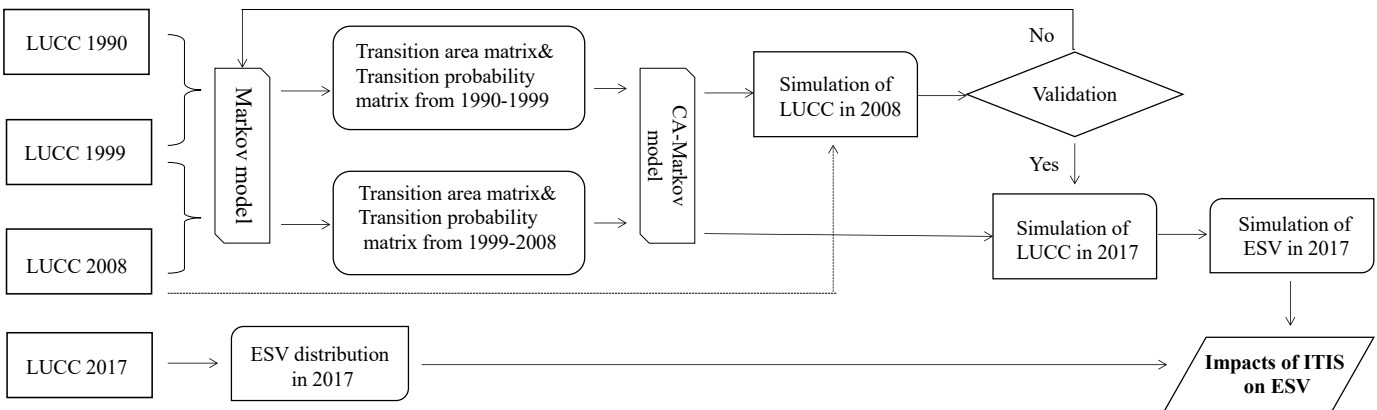

**Figure 2.** Technical roadmap.

### 4.4. Ecosystem Service Value Estimation

In this paper, using the research results of Xie et al. [11] and taking into account the situation on Hainan Island and data availability, the following revisions were made to the ESV coefficients of land classes in terms of socio-economic aspects.

#### 4.4.1. Revisions Based on Food Prices

Xie et al. [11] defined one standard ecosystem ecological service value equivalent factor as the economic value of the annual natural grain yield of a 1 hm$^2$ national average yield farmland, which is equal to one-seventh of the market value of the average grain yield in the study area in that year. Based on this, the value of one equivalent factor can be corrected according to the following equation:

$$E_t = \frac{1}{7}\frac{T_t}{M_t} \tag{1}$$

where $E_t$ is the economic value of food production services provided per unit area of farmland ecosystem in year $t$; $T_t$ is the total food production value of Hainan Island in year $t$ at constant 2017 prices; and $M_t$ is the food cultivation area of Hainan Island in year $t$.

#### 4.4.2. Social Development Coefficient Correction

The value of ecosystem functions is influenced by people's willingness to pay for ecosystem functions and services, which needs to be corrected by the social development coefficient [52,53]. In this paper, the Peel growth curve (S-curve) model is used to identify the social development coefficient, and the ratio of the study area to the national average social development coefficient is used as the correction coefficient, taking into account the urban and rural development levels. The specific calculation formula is as follows:

$$Q = \frac{1}{1 + ae^{-b(\frac{1}{E_n}-3)}} \tag{2}$$

$$Q' = Q_1 \times P_1 + Q_2 \times P_2 \tag{3}$$

where $Q$ is the coefficient of social development stage related to willingness to pay, generally a and b constant is 1; $E_n$ is the Engel coefficient; and $e$ is the natural constant. $Q_1$ denotes the urban social development coefficient; $Q_2$ denotes the rural social development coefficient formula; $Q'$ denotes the overall social development factor; and $P_1$ and $P_2$ refer to the proportion of urban and rural population in the total population, respectively.

4.4.3. Ecosystem Service Value Assessment Model

The value of ecosystem services for different land types on Hainan Island at each study time point was obtained from the table of ecosystem service equivalent factors per unit area and the modified value equivalents with the following equation:

$$ESV_t = \frac{\text{Value}_t}{\text{CPI}_t} \times 100 \tag{4}$$

$$\text{Value}_t = \sum_{i=1}^{i} S_t A_i E_{ij} E_t j = 1, 2, 3 \ldots 11 \tag{5}$$

where *i* is the ecosystem type; *j* is the ecosystem service function; and *t* represents the year. $S_t$ is the social development correction coefficient in year *t*; $E_t$ is the economic value of ecosystem services per unit area in yuan/ha in year *t*; $E_{ij}$ is the equivalent value of the *j* ecological service provided by the *i* land-use type; $E_t E_{ij}$ is the value of ecosystem services per unit area in yuan/ha for the *i* land-use type; and $A_i$ is the area in hectares for the *i* land-use type; CPI is an index of inflation for Hainan province.

## 5. Results and Analysis

*5.1. Analysis of the Spatial and Temporal Evolution of ESV in Hainan Island*

The ESV of Hainan Island was 33.88, 56.045, and 50.417 billion yuan in 1999, 2008, and 2017, respectively, with an overall trend of increasing and then decreasing (Figure 3).

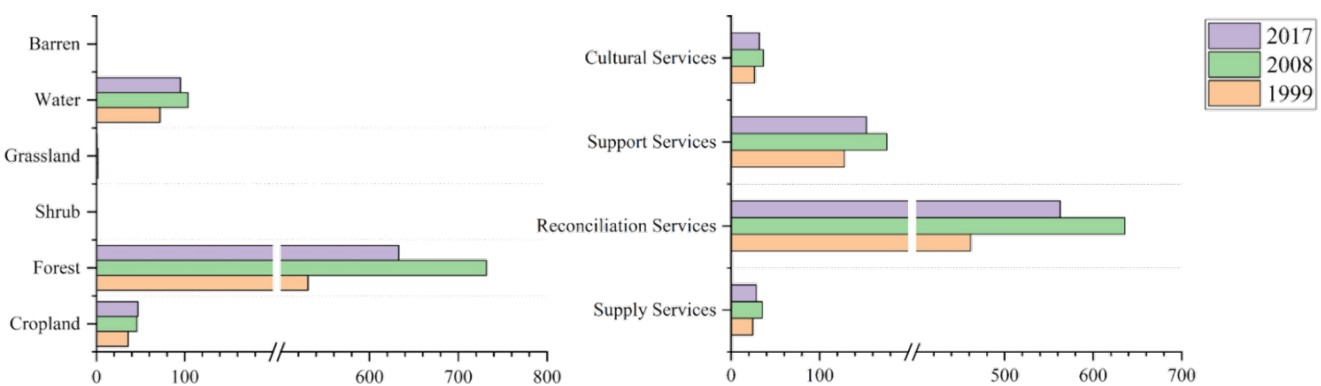

**Figure 3.** Temporal evolution of ESV in Hainan Island from 1999 to 2017.

In terms of spatial distribution, from the raster scale, the ESV of Hainan Island was high in the central region and low in the surrounding area, increasing from the coast inland, and the overall spatial pattern is relatively stable during the study period. The ESV of each city and county ranges from 0.949 billion to 5.303 billion yuan, with Danzhou, Qiongzhong, and Ledong having higher ESVs and Lingshui County having the lowest ESVs in all three study years. From 1999 to 2008, the ESV of each city and county increased, with the largest change occurring in Danzhou City, which increased by about 2.2 billion yuan during the decade. Wanning had the smallest rise, with an increase of about 600 million yuan in ESV. During the implementation of the ITIS in 2008–2017, the value of ecosystem services in each city and county was declining, with Haikou experiencing the largest decline, that of about 0.8 billion yuan, followed by Wenchang and Ledong. The ESV of Wanning declined the least, by only about 210 million yuan (Figure 4).

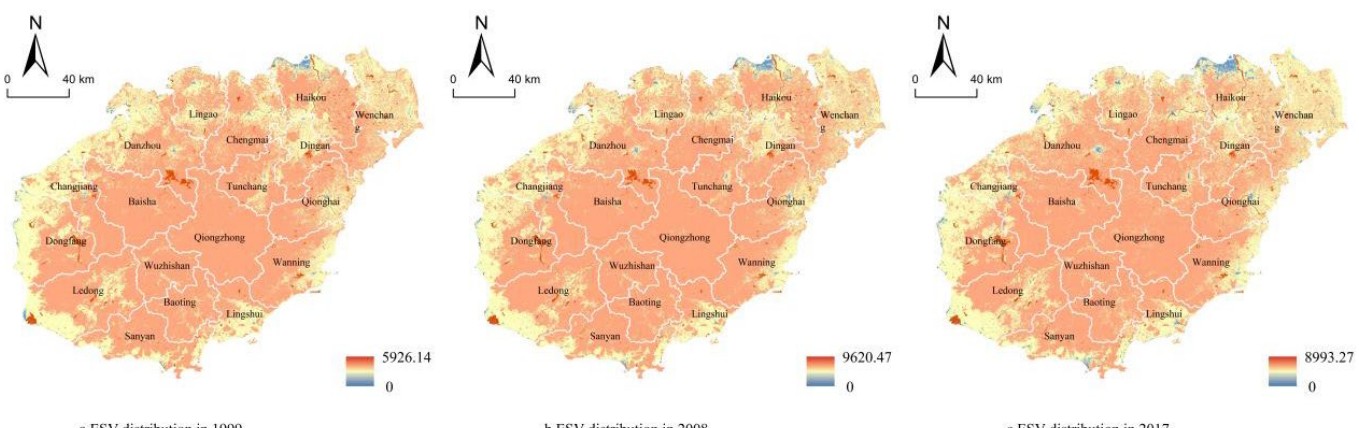

**Figure 4.** Spatial evolution of ESV in Hainan Island from 1999 to 2017.

The structure of ESV land-use types in Hainan Island was relatively stable during 1999–2017; from largest to smallest, the land-use types were ordered as follows: forest land, water, farmland, grassland, shrubland, and wasteland. Forest land accounted for 80% of the ESV. The proportion of water in the land-use types of Hainan Island is small—only about 2%—but it provides the second highest ESV value share, which is mainly related to the water having a large ESV coefficient; therefore, its ecosystem service capacity is very strong. The ESV of forest land increased the most between 1999 and 2008, followed by water and farmland. During 2008–2017, the ESV of forest land decreased the most, followed by watershed and grassland, and the ESV of farmland increased during this period.

Next we examine the different service functions. Structurally, regulating services play a major role in the primary service type, accounting for about 72% of the total service value, followed by supporting services, aesthetic landscape, and supply services. The secondary service types are mainly hydrological regulation and climate regulation, and water resources supply is in an inferior position among all ecosystem service functions of Hainan Island. In terms of variation, the ESV of all ecosystem service types except water supply tended to increase from 1999 to 2008, while all tended to decrease from 2008 to 2017. The largest change in regulating services was 15.9 billion dollars during the 20 years studied.

*5.2. Analysis of 2017 ESV Simulation Results for Hainan Island under Counterfactual Scenarios*

The simulation accuracy verification results show that the Kappa coefficient between the simulated and actual 2008 land-use cover data in the study area is 0.88, indicating that the results of the simulated data in 2008 are in high agreement with the true situation. Therefore, the 2017 land-use cover data obtained from further simulations based on this are highly credible and can be used for comparative analysis under counterfactual scenarios. The simulation results show that the total simulated ESV in 2017 is 54.19 billion yuan under the counterfactual scenario, assuming that the ITIS is not implemented (Figure 5).

In terms of spatial distribution, the 2017 ESV simulation results for Hainan Island are high in the central region and low in surrounding areas. From the scale of each city and county, the ESV ranges from 1.431 to 4.908 billion yuan, with the highest simulated ESV value of 4.908 billion yuan found in the central Qiongzhong region, followed by Danzhou and Ledong, with 4.842 billion yuan and 4.171 billion yuan, respectively. The lowest is still Lingshui, with an ESV of 1.431 billion yuan.

In terms of different land-use types, under the scenario in which the policy is not implemented, the contribution structure of ESV land types in Hainan Island in 2017 is still forest land, water, farmland, grassland, shrubland, and wasteland, of which the contribution value of forest land increased to 41.431 billion RMB and the contribution value of wasteland is very small.

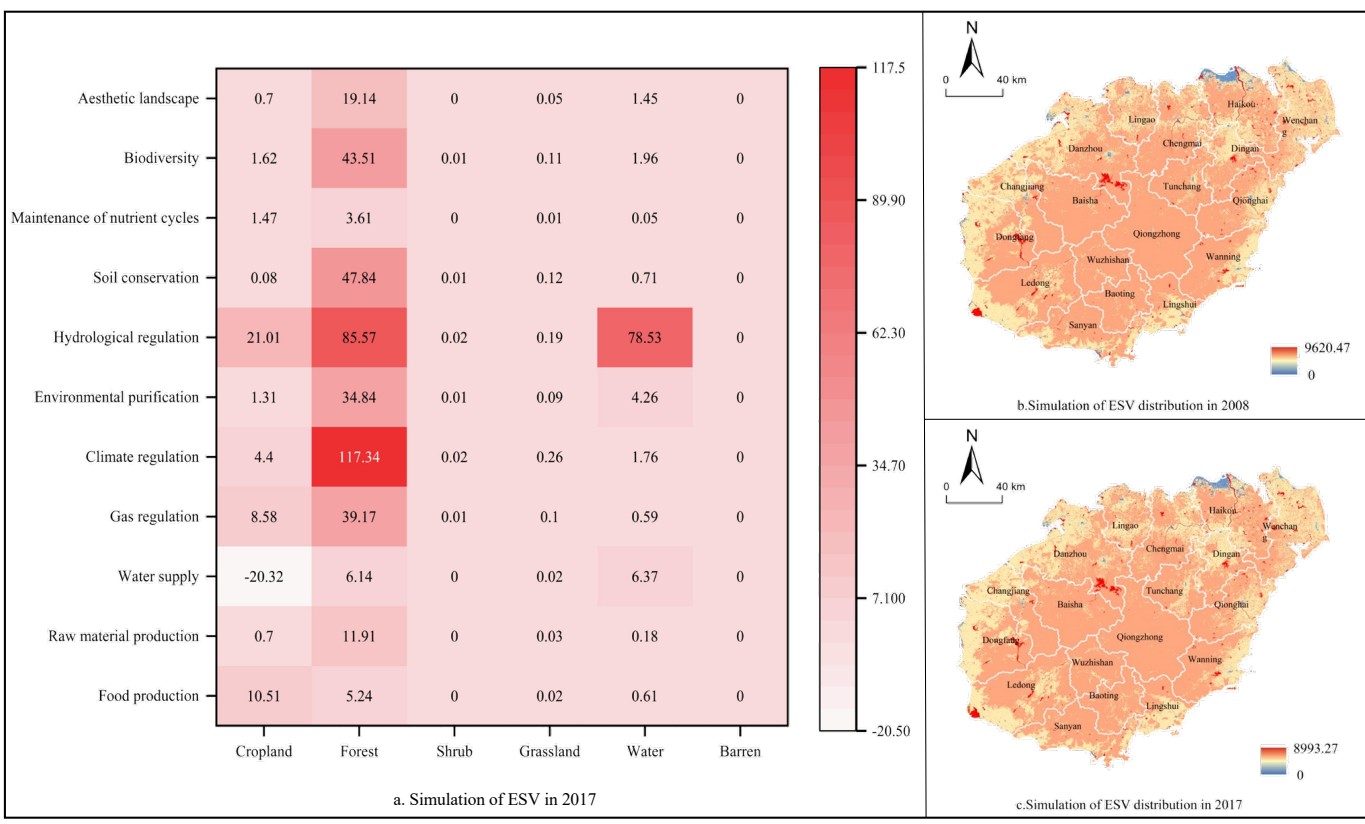

**Figure 5.** ESV model simulation results, 2017.

In terms of different service functions, the contribution is, from highest to lowest: regulation services, support services, cultural services, and supply services, with the ESV provided being 39.806, 1.011, 2.140, and 2.1337 billion yuan, respectively. In addition, hydrological regulation dominates, providing 18.532 billion yuan of ESV, and water supply occupies an inferior position, providing a value of −0.78 billion yuan of ESV.

Overall, in terms of the distribution of land use types, the total area of agricultural land and impervious surface in the scenario of Hainan Island without the implementation of the international tourism island strategy is smaller than the actual situation, and the degree of expansion is reduced by 17,070.57 hectares and 6474.87 hectares, respectively; while the total area of woodland, shrubland, grassland, and water will be larger than the actual situation, which shows that if the international tourism island strategy is not implemented, the demand for construction land will be greatly reduced and more ecological land will be preserved. In terms of the distribution of ecosystem service values, the simulated pattern of ESV on Hainan Island in 2017 is generally consistent with the true scenario. However, the area of impervious surface expansion in areas with low ESV is substantially reduced compared to reality, while more woodlands, grasslands, and watersheds are preserved in areas with high ESV (Figures 4c and 5c). The overall ecological situation in the simulation is better than the actual scenario.

*5.3. The net Spatial and Temporal Impact of the ITIS on ESV in a Counterfactual Analysis Framework*

Under the counterfactual analysis framework, comparing the simulated and actual scenarios of the ESV of Hainan Island in 2017 shows that the implementation of the ITIS strategy reduced the ESV of Hainan Island by 3.773 billion yuan. This means that the implementation of the ITIS would still result in an island-wide reduction in ESV if the changes brought about by Hainan Island's own socio-economic development are excluded.

Comparing the spatial situation of ESV with and without the implementation of the international tourism island policy, it is clear that the degree of spatial impact of the ITIS

on the ESV of Hainan Island shows a higher degree of coastal change than inland. Except for the unchanged raster, the implementation of the ITIS significantly reduced the ESV in the northeast and southwest areas of Hainan Island, while the increased ESV is mainly distributed in the northwest and north of the forest and farmland junction zone.

There is heterogeneity in the ESV of different cities and counties affected by the ITIS, and about two-thirds of the cities and counties suffered a loss in ESV due to the strategy. Among them, the ITIS had the greatest degree of impact on Haikou city, which reduced its ESV by 478 million yuan; it had the least degree of impact on Qionghai city, which increased by 3 million yuan. In addition, the ESV of Lingshui, Wuzhishan, Baoting, Tunchang, Dongfang, Chengmai, Sanya, Qiongzhong, Ding'an, Ledong, Wanning and Wenchang decreased to different degrees, while the ESV of Danzhou, Changjiang, Lingao, and Baisha increased (Figure 6).

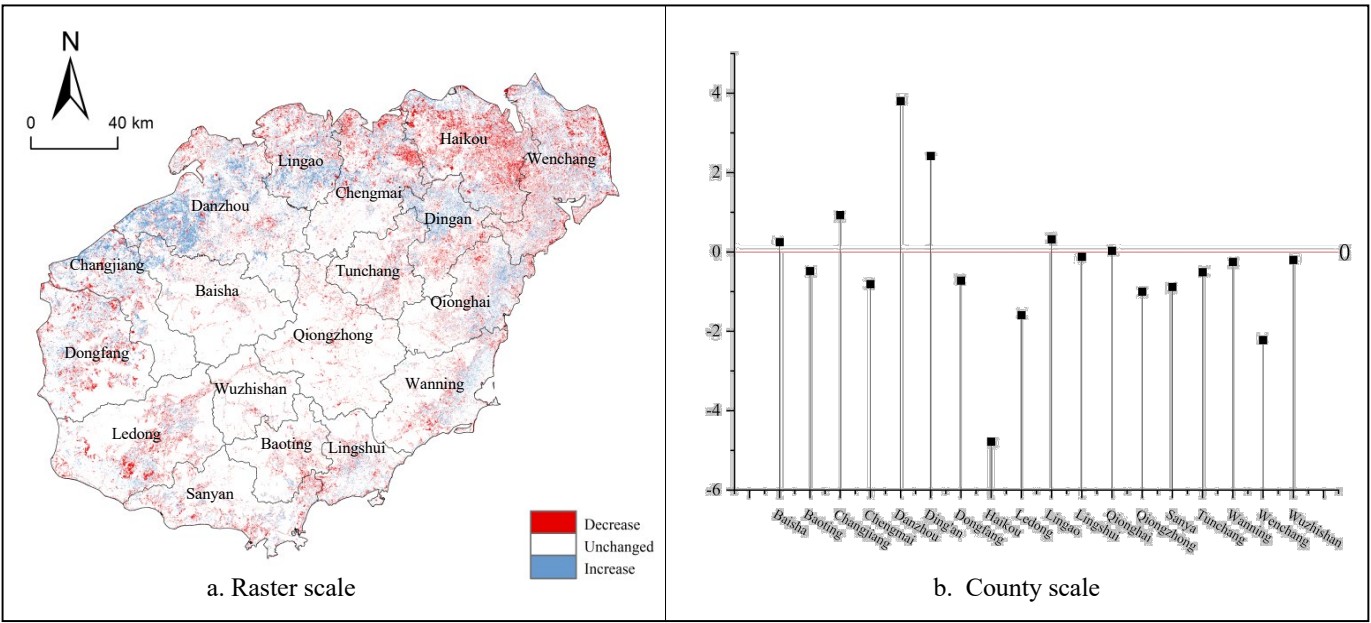

**Figure 6.** Net spatio-temporal impact of ESV for the ITIS.

## 6. Discussion

### 6.1. Analysis of the Ecological Impact of the International Tourism Island Strategy on Hainan Island and its Causes

Overall, the implementation of the international tourism island strategy has led to a decline in the ecological situation of Hainan Island, with a decrease in ESV value of 3773 million yuan, and the degree of impact is characterized by a higher degree of coastal than inland, with the most pronounced decrease in ESV in the northeastern and southwestern coastal areas of Hainan Island. In addition, compared to 1999–2008, the value of ecosystem services on Hainan Island showed a sharp decrease from 2008–2017.

For the ecological impact of the international travel island policy, this paper analyzes the causes from two aspects: the expansion of construction land and the occupation of ecological land.

Compared to 1999–2008, the value of ecosystem services on Hainan Island showed a sharp decrease from 2008 to 2017, which is closely related to the implementation of ITIS. The expansion of construction land, the low-value sector of ESV, is expanding. During the construction of the international tourist island, the expansion of construction land was mainly driven by three factors: industry, infrastructure, and population. First, the area has witnessed the cultivation of tourism and real-estate-based industries on Hainan Island, with leisure and vacation projects, South China Sea resource development, and service bases landing on the island. Second, the island engaged in the construction of public services and

infrastructure, such as tourist attractions, highways, railroads, and airports; the completion of the interconnected traffic network described as "one vertical, three horizontal, and four rings"; and the traffic network called the "two-hour economic traffic circle". Finally, the introduction and settlement policies and the emphasis on the education and training of professional tourism talents have attracted many new employees to settle in Hainan, driving the province's housing demand. Based on this, the ITIS resulted in a net increase of 518.14 km$^2$ of construction land on Hainan Island between 2008 and 2017, with a loss of ESV. In addition, the agglomeration of construction land will also have certain negative external effects on the surrounding area, thus influencing the regional ecological situation under other land-use patterns and affecting the overall ESV of Hainan Island.

Furthermore, the occupation of ecological land, the high-value area of ESV in Hainan Island, is shrinking. The land area of the island is limited and consists mainly of forest land, agricultural land, and construction land, and the spatial distribution pattern of ESV is high in the central region and low in the surrounding areas. With the continuous promotion of the ITIS, coupled with the low importance attached to ecological land and the perfection of existing policies, construction land is gradually extended to the central section of the island, and ecological land is continuously occupied. This means that the main provision area of ESV in Hainan Island (i.e., the central concentrated forest land) shrinks and ESV decreases.

## 6.2. Differences in the Assessment Results: Traditional Assessment Thinking vs. the Counterfactual Analysis Framework

Overall, the assessment method that compares the differences between before and after the implementation of the ITIS strategy overestimates the impact of the strategy's implementation on the value of ecosystem services on Hainan Island. The degree of ecological impact of the ITIS policy is assessed at 4.724 billion yuan higher than that obtained through the traditional approach. It is thus clear that, despite the absence of an international tourism island policy, the increasing level of development on Hainan Island will still have an impact on the island's land use and ecosystem as the economy and society continue to develop.

Spatially, the reduction of ESV in the vast majority of cities and counties under the traditional assessment approach is greater than the value under the counterfactual assessment philosophy. This indicates that even without the implementation of the international tourism island construction strategy, each province has certain potential for its own development, thus influencing the evolution of the ecosystem, which is accelerated by the implementation of the ITIS. Cities and counties such as Haikou, Danzhou, and Wenchang have good economic, educational, demographic, and cultural aerospace infrastructures, which will result in the loss of ESV in the process of their own development, even if they do not implement the ITIS. For example, the forest land in Qiongzhong occupies 99% of the total area of land, so most of the land occupied in the process of development is forest land, which will result in a sharp decrease of ESV in this region (Figure 7).

## 6.3. Shortcomings and Prospects
### Policy Insights

The implementation of the ITIS has played an important role in the economic and social development of Hainan Island, but combined with the research in this paper, it is evident that the strategy has also caused a decline in the overall ecological situation, giving rise to a series of environmental problems, such as land-use changes and human–land tension. In 2018, Hainan Island increased the pace of free trade port construction and, at the same time, as a national ecological civilization pilot area, Hainan Island is significant. Combining the above results, the following policy recommendations are proposed.

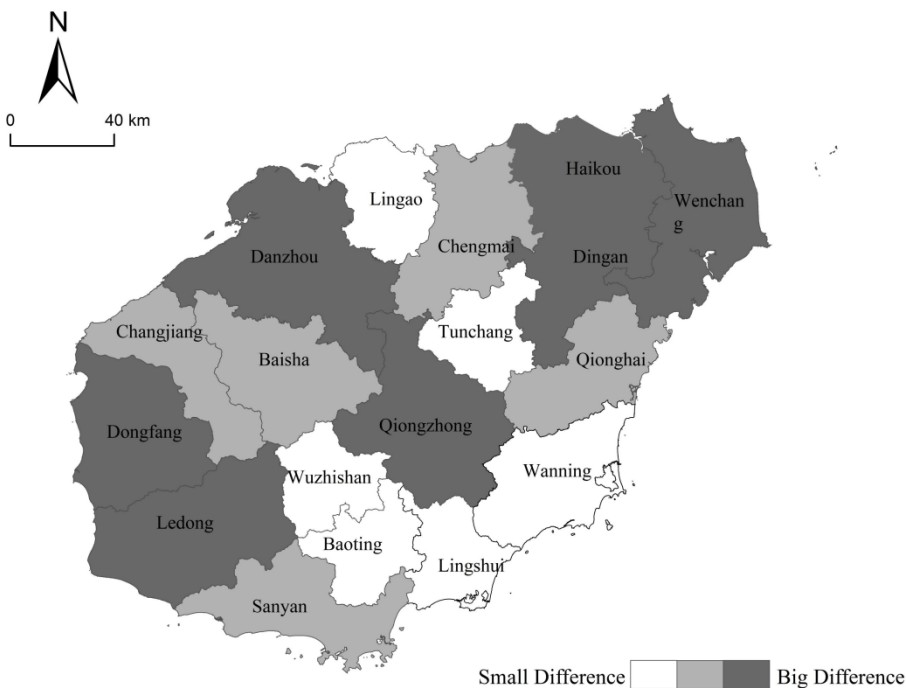

**Figure 7.** Comparison of ESV differences between cities and counties on Hainan Island under two assessment ideas.

First, protection of ecological land. Combining the spatial and temporal evolution trend of ESV in Hainan Island from 1999 to 2017, the most widely distributed forest land and water with high ecological coefficients occupy more than 90% of the total ESV in Hainan; currently, however, these two land types are decreasing in area. Therefore, it is necessary to develop a refined and localized land-use plan to enhance their protection. In addition, the supply service of ESV in Hainan Island has been negative during the two decades studied in this paper, so it is necessary to protect the watershed and forest land and to be alert to the negative impacts of the expansion of agricultural land on the supply service.

Second, use of wasteland. During 1999–2017, the transfer of wasteland to construction land on Hainan Island was minimal, with only 258.03 ha of wasteland converted to construction land; instead, the expansion of construction land occupied more areas with high ESV. This shows that a large amount of wasteland is in an idle state, which also provides the possibility of more efficient land use and the increase of ESV on Hainan Island.

Third, adaptation to local conditions. The forest land of Hainan Island is mainly distributed in the central inland mountainous area, which is also the main area of ESV provision. Meanwhile, the coastal areas, including Haikou and Sanya City, are important for the economic and social development of Hainan Island; they are also areas that experienced a large reduction of ESV after the implementation of the ITIS. Therefore, zoning can be used to take advantage of the strengths of different regions and to maintain the overall balance of ESV on Hainan Island. However, in line with the Chinese modernization value stating that "the fruits of development are shared by the people", there is a need to prevent uncoordinated development levels on the island that would widen the gap in the standard of living of residents. This can be achieved by linking land increases and decreases, balancing land occupation and replenishment, and implementing corresponding ecological compensation measures, in line with the national strategy of free trade port construction, to promote the coordinated and sustainable development of the island's economy, society, and ecosystem.

## 7. Conclusions

This paper uses a counterfactual analysis framework and simulates and calculates ESV using the CA-Markov model as well as the equivalent factor method to assess the spatial and temporal evolution of ESV on Hainan Island brought about by the implementation of the ITIS at the land class, raster, and city–county scales. We reach several conclusions.

First, the spatial and temporal distribution characteristics of ESV in Hainan Island over the years. The ESV of Hainan Island was 33.89 billion, 56.045 billion and 50.417 billion yuan in 1999, 2008 and 2017, respectively. The structure of the land type contribution is forest land, watershed, farmland, grassland, shrubland, and wasteland, in that order, with the greatest degree of ESV change occurring in forest lands and watersheds. The primary service type plays a major role in regulating services, followed by supporting services, aesthetic landscape, and supply services. The secondary service types are mainly hydrological and climatic regulation, and water supply is in an inferior position in the overall ecosystem service function of Hainan Island. Spatially, the ESV of Hainan Island is high at the center and low in the surrounding areas, with higher ESV values found in Danzhou, Qiongzhong, and Ledong; the largest ESV decline was in Haikou—about 1.3 billion yuan.

Second, there are several notable spatial and temporal distribution characteristics of ESV in Hainan Island under the unimplemented policy scenario. In 2017, under the simulated scenario of the unimplemented strategy, the total ESV in Hainan Island was 54.19 billion yuan, and the main ESV contributing land types remained forest land and watersheds. Spatially there was little change in the distribution of ESV compared to previous years.

Third, we examine the net effect of policy implementation on ESV in time and space. The implementation of the ITIS reduced ESV by 3.773 billion yuan. The degree of spatial impact of the strategy on ESV showed a higher degree on the coasts than inland. There was a significant decrease in ESV in the northeast and southwest of Hainan Island, and an increase in ESV at the junction of forest and farmland, such as in the northwest and the north. The policy had the greatest degree of impact on Haikou. In addition, the traditional assessment method overestimates the impact of the ITIS policy on the value of ecosystem services on Hainan Island, and the ecological impact of the policy is assessed to be 4.724 billion yuan higher than that estimated by the traditional method.

There are certain shortcomings and directions for further research in this paper. First, the expansion of construction land will have certain negative effects on water harvesting and wastewater treatment, but this paper does not include the role of construction land because a unified assessment standard has not yet been formed. Second, based on the accessibility of data, this paper corrects the ecosystem service value equivalent per unit area using the food production and social development coefficient, and analyzes the land-use type as the scale. However, the natural and economic and social conditions of different plots in the same land-use type also differ, resulting in some deviations in the total ESVs. In the future, multi-source spatial data can be adopted to make relevant corrections for different parcels. Finally, although the counterfactual analysis framework in this paper excludes the development potential of the region itself, the construction of high-speed rail, the implementation of small-scale development policies, and the radiation of other regions in space will also have some influence on the results, and it is worth considering how to better reduce the influence of other confounding factors.

**Author Contributions:** Writing—original manuscript preparation, M.G.; Writing—review and editing, C.X. All authors have read and agreed to the published version of the manuscript.

**Funding:** This research was funded by the Ministry of Education in China Key Projects of Philosophy and Social Sciences Research: 20JZD013 & the National Natural Science Foundation of China: 72004049 & the Ministry of Education in China Liberal Arts and Social Sciences Foundation: 20XJCZH009 & the High-level Talents Program of Hainan Provincial Natural Science Foundation of China: 2019RC122, 2019RC025 & the Hainan Provincial Natural Science Foundation of China:

720QN241 & the Planning Research Program of Hainan Provincial Philosophy and Social Science Foundation of China: JD(ZC)19-40.

**Institutional Review Board Statement:** Not applicable.

**Informed Consent Statement:** Not applicable.

**Data Availability Statement:** Not applicable.

**Conflicts of Interest:** The authors declare no conflict of interest.

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
