# Peer review of "The Net Spatio-Temporal Impact of the International Tourism Is-Land Strategy on the Ecosystem Service Value of Hainan Island: A Counterfactual Analysis"

_land, doi:10.3390/land11101694_

Round 1
Reviewer 1 Report
Review of the manuscript „The net spatio-temporal impact of the international tourism is land strategy on the ecosystem service value of Hainan Island: A counterfactual analysis”
The authors clearly state the purpose of the article. Generally, the article is well written, concepts are clearly defined. The structure of the text correct.
Please improve the quality of the maps, they should be larger and legible (especially Figure 2), and carefully check the English language.
Please include suggestions for further studies in the discussion section.
Text formatting should be improved (page 6, line 237-255).
The article fits the scope of Land Journal. The topic of the article is interesting and up-to-date.
Reviewer 3 Report
It is interesting to use a counterfactual analysis framework to analyze the impact of policy on the ecosystem service value of Hainan Island. The research results would provide some inspiration to the readers. Here are some suggestions:
Figure 2, The technical roadmap has not well reflect the technical steps, besides, the pictures included are not clear and duplicated with those in other parts. I suggest containing more details and describing with keywords and flow charts.
How is the CA-Markov model like in this research? Is it an open accessed model or a newly developed model, and what is the input data?
Section 4.2, considering the most critical factor to affect the results of ESV is the difference between the actual l land-use cover and the simulated land-use cover, it is better to first describe the difference matrix of each land use type between these two data sets.
Figure 6, Define the quantitative category of High and low.
Cultural value and recreation value are also important aspects of ecosystem services, how are these factors reflected in your research?
Reviewer 4 Report
Thank you for allowing me to review this interesting research. My recommendations to improve the paper are:
1. Rewrite the abstract to be clearer and understandable
2. Split the Introduction section into 1. Introduction and 2. Literature review. In the Introduction insert/add hypotheses, research question, motivation, goal(s) and objective(s) of the study. In the Literature review section add 20 most recent citations on the research field.
3. What are the numbers in lines 186 and 187?
4. Do copy editing. There are some mistakes.
5. I didn't get the idea of probabilities using the Markov chain for the variables used and the length of the period. There are too many different years. Please be more concise and precise, about which years, and which variables are tested in the model. Moreover, you wrote inflation to avoid the effect of price rise, but in the end, the index of food prices is used. This is not inflation. On top of that in the equation PPI is used. Please use the most recent literature to see how we deal with deflation of prices.
6. It would be beneficial to answer your research question(s)/hypotheses in the discussion section. Moreover, the limitations and delimitations should be moved from the discussion section to the conclusions.
7. Figure 3 is not readable. Split it.
8. What is KMB? Line 343.
Good luck.
Round 2
Reviewer 4 Report
Dear authors.
Thank you for your revision. However, I still do not see this paper as suitable for publishing. You need to:
1. rewrite the abstract in a clearer version. At the moment it means shorter sentences.
2. The introduction section needs to be split into at least two paragraphs. Did you add your research question?
3. line 46: "To achieve these goals," Which goals? Please highlight them.
4. in equation 3 what is the meaning of "`" on Q (Q`)?
5. line 285, "CPI is the domestic demand index". As far as I know, CPI is the consumer price index and not the domestic demand index
6. You did not add any new citations (literature). Do that appropriately. I previously asked for 20 of them.
7. Copy editing still needs to be done: for example: "Overall,In terms o", line 363.
8. line 375: can you explain where I can see (Figure 4c and Figure 5?
Thank you and good luck.
Round 3
Reviewer 4 Report
Dear authors,
Authors have now implemented almost all the open issues.
CPI is an index of inflation for Hainan province is incorrect. Revise to: CPI is the consumer price index indicating the inflation in the Hainan province.
Good luck.